# Deep mtDNA Sequence Divergences and Possible Species Radiation of Whip Spiders (Arachnida, Amblypygi, Phrynidae, *Phrynus/Paraphrynus*) among Caribbean Oceanic and Cave Islands

Ingi Agnarsson [1,2,*] , Jonathan A. Coddington [2] , Laura Caicedo-Quiroga [2,3,4] , Laura J. May-Collado [5,6] and Snæbjörn Pálsson [1]

1 Faculty of Life and Environmental Sciences, University of Iceland, Sturlugata 7, 102 Reykjavik, Iceland
2 Department of Entomology, National Museum of Natural History, Smithsonian Institution, NHB-105, P.O. Box 37012, Washington, DC 20013, USA
3 Walter Reed Biosystematics Unit (WRBU), Smithsonian Institution Museum Support Center, Suitland, MD 20746, USA
4 One Health Branch, Walter Reed Army Institute of Research (WRAIR), Silver Spring, MD 20910, USA
5 Department of Biology, College of Arts and Sciences, University of Vermont, 109 Carrigan Drive, Burlington, VT 05401, USA
6 Smithsonian Tropical Research Institute, Apartado 0843-03092, Balboa, Panama
* Correspondence: iagnarsson@hi.is

**Abstract:** Islands—whether classic oceanic islands or habitat islands such as isolated thermal vents, mountain tops, or caves—often promote the diversification of lineages that colonize them. We examined CO1 mtDNA sequence divergences within the tailless whip spider genus *Phrynus* Lamarck, 1809 (Amblypygi: Phrynidae) among oceanic islands and among cave 'islands´ distributed across the Caribbean archipelago and on the continental mainland. The significance of this study lies in the extensive taxon sampling of a supposedly depauperate lineage (considering its age), over a large proportion of its geographical range, and the discovery of deep mtDNA sequence divergences. We sampled thousands of specimens—and sequenced 544, including six outgroup species—across 173 localities on 17 islands (135 localities) and five countries on the North to South American mainland (38 localities), including a total of 63 caves. Classical taxonomy identified ten named *Phrynus* and two *Paraphrynus* Moreno, 1940 species. *Paraphrynus* seems to be paraphyletic and nested in *Phrynus*. Uncorrected genetic distances within named species and among morphological species ranged up to 15% and 19%, respectively. Geographic distances explained a significant portion of genetic distances on islands (19%, among both subterranean and epigean specimens), and for epigean specimens on the mainland (27%). Species delimitation analyses indicated that the 12 named species harbored from 66 to well over 100 putative species. The highest number of species was indicated by the GMYC method (114 species) while the Bayesian Poisson tree processes (bPTP) and the BP&P relying on the Markov chain Monte Carlo Bayesian Phylogenetic model estimated an upper level of 110 species. On the other hand, the recently recommended and relatively conservative distance-based (phylogeny free) ASAP model has the greatest support for 73 species. In either case, nearly all putative species are tightly limited to a single locality, often a small cave system, and sometimes to the surrounding epigean area. Caribbean *Phrynus* diversity has likely been vastly underestimated, likely due to both morphological crypsis and the ignorance of Caribbean cave fauna. Although mtDNA sequences can suggest species limits, nuclear DNA sequencing and detailed morphological research are necessary to corroborate them and explore whether this phenomenon constitutes species radiation or perhaps just mtDNA divergences as a consequence of, for example, stationary females and actively dispersing males.

**Keywords:** barcoding; biodiversity hotspot; cavernicolar; cryptic radiation; habitat islands; speciation; endemism

## 1. Introduction

The Caribbean archipelago is a recognized biodiversity hotspot due to species richness and especially high, often single island, endemism [1–4]. Speciation often occurs within the larger islands among habitats or across barriers such as mountain ranges. In the Caribbean, the rich cave systems aptly called habitat 'islands' [5–10] are another understudied dimension of diversity. Cave-liking fauna often displays limited movement among caves or cave systems, resulting in relatively small geographic population ranges and a lack of gene flow and higher degrees of micro-endemism compared to epigean habitats [9,11,12]. The thousands of biotically underexplored subterranean systems in the Caribbean have likely promoted diversity as 'islands within islands' [10]. Cave-liking taxa that also live in epigean habitats may speciate across oceanic barriers among islands, across habitat barriers such as mountain ranges, and at the smallest scale among cave systems within otherwise homogenous regions [5,8], including various arachnids [9,10,13–15].

Phrynidae (Amblypygi) (Figure 1) are ideal to test the hypothesis that caves may promote speciation, as they are found both in epigean habitats and in practically all sampled Caribbean caves. They are also thought to be dispersal-limited by life-long site fidelity to single trees or caves [10,16,17]. The order is by any measure a "depauperate" lineage: Estimated to be up to 400 my old but containing only 262 extant described species. *Phrynus* itself reiterates this pattern: ~130 my, 36 described species [18,19]. This study presents the largest sample to date from nearly the entire range of the genus. We explored mtDNA sequence divergences and putative species radiation in tailless whip spiders based on genetic sampling from 544 specimens across the Caribbean islands and on neighboring continental landmasses (Figure 1, Table 1). Most of these specimens (464) came from approximately 12 morphological species of *Phrynus* and *Paraphrynus* (Table 1). We used various methodologies to estimate sequence divergences and molecular distances and exploring four recently developed species delimitation methods to reveal the CO1 mtDNA genetic structure and species of *Phrynus* among islands, habitats, and caves. We discuss the evidence for species radiation in this lineage and highlight ways to discriminate between rapid mtDNA divergences within species and cryptic species radiation.

**Table 1.** Samples, number of specimens per country (*n*), sampled from caves, and morphological, ASAP and Poisson tree processes (PTP—a phylogenetic tree-based method) species per country. *: Lesser Antilles.

| Country | Code | *n* | Number of Specimens in Caves | Number of Morpho-Species | Number of ASAP Species | Number of bPTP Species |
|---|---|---|---|---|---|---|
| Antigua * | AN | 3 | 2 | 2 | 2 | 3 |
| Barbados * | BS | 8 | 6 | 1 | 1 | 1 |
| Barbuda * | BA | 3 | 3 | 1 | 1 | 1 |
| Guatemala | BE | 3 | 3 | 1 | 1 | 1 |
| Colombia | CO | 54 | 1 | 3 | 7 | 8 |
| Costa Rica | CR | 14 | 4 | 2 | 3 | 3 |
| Cuba | CU | 34 | 7 | 6 | 11 | 13 |
| Dominica * | DO | 13 | 0 | 1 | 3 | 4 |
| Dominican Republic | DR | 53 | 22 | 5 | 13 | 21 |
| Guadeloupe * | GU | 1 | 0 | 1 | 1 | 1 |
| Jamaica | JA | 42 | 29 | 3 | 8 | 8 |
| Mexico | MX | 54 | 21 | 3 | 15 | 19 |
| Monserrat * | MO | 2 | 0 | 1 | 1 | 1 |
| Puerto Rico | PR | 184 | 139 | 5 | 7 | 20 |

**Table 1.** *Cont.*

| Country | Code | *n* | Number of Specimens in Caves | Number of Morpho-Species | Number of ASAP Species | Number of bPTP Species |
|---|---|---|---|---|---|---|
| Saba * | SA | 3 | 5 | 1 | 1 | 1 |
| St. Barts * | SB | 3 | 0 | 1 | 1 | 1 |
| St. Eustatius * | SE | 1 | 0 | 1 | 1 | 1 |
| St. Kitts * | SK | 12 | 3 | 1 | 1 | 1 |
| St. Martin * | SM | 2 | 0 | 1 | 2 | 1 |
| St. Vincent * | SV | 11 | 1 | 1 | 2 | 3 |
| Turks and Caicos | TC | 19 | 9 | 1 | 1 | 1 |
| USA, Florida | FL | 23 | 0 | 1 | 2 | 2 |

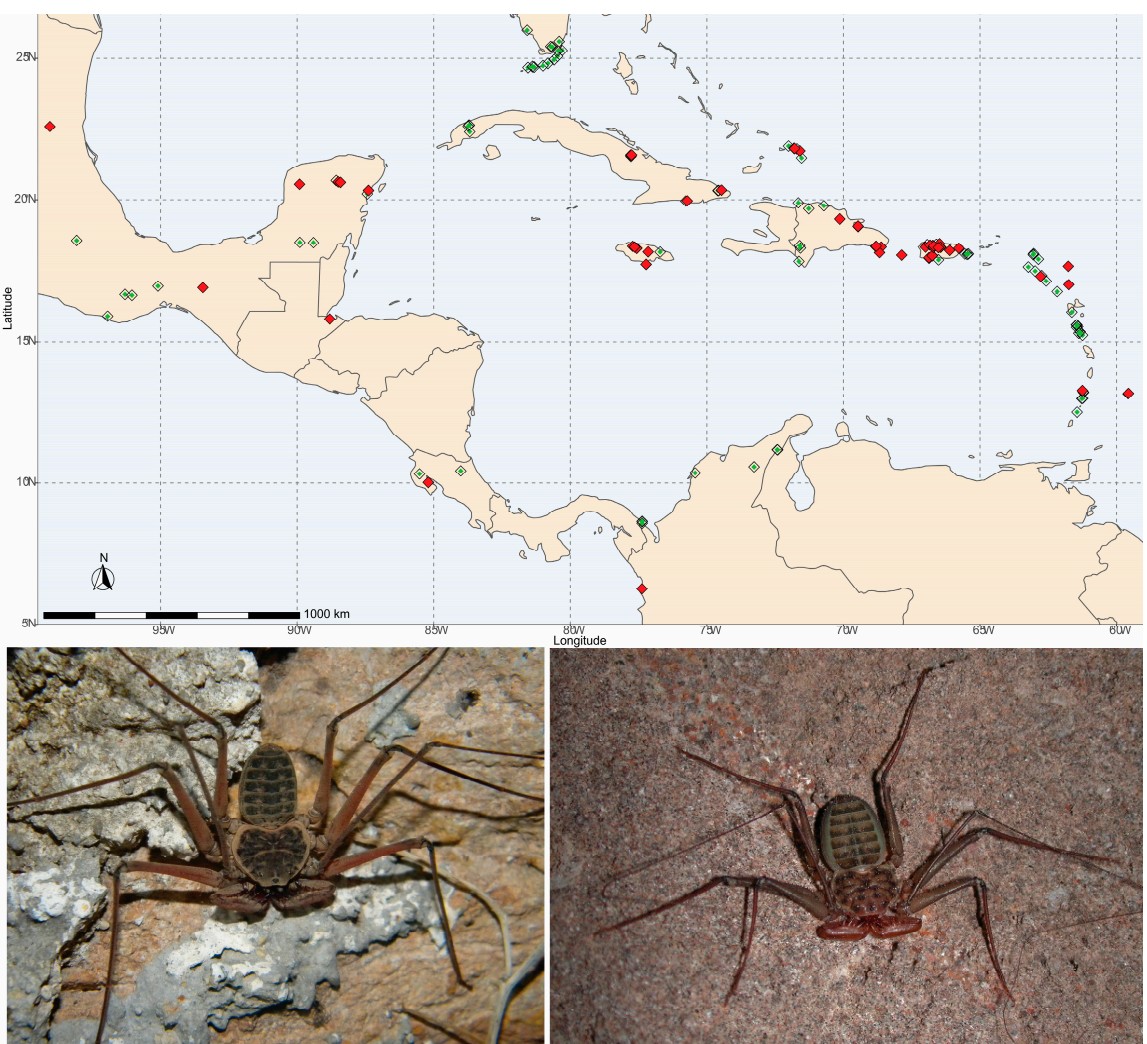

**Figure 1.** (**Above**) Sample sites in the Caribbean where green and red diamonds show sampling in epigean and cave habitats, respectively. Thicker lines around diamonds indicate multiple nearby collection localities. (**Below**) Photographs of *P. longipes* ((**left**), By Maximilian Paradiz from Amsterdam, Netherlands–*Phrynus*, CC BY 2.0, https://commons.wikimedia.org/w/index.php?curid=64070641 and *Paraphrynus mexicanus* ((**right**), by Marshal Hedin from Wikipedia commons (both accessed on 15 February 2023)).

Species delimitation analyses (or DNA barcoding sensu Hebert et al. [20]) based on the mtDNA COI gene is a common method of species discovery and provides a preliminary estimate of species richness, especially in systems where morphological evidence is sparse or unreliable (reviewed in [21] for spider studies, see also [22–24]). We highlight four approaches that differ in how the evidence is assessed. The Generalized Mixed Yule Coalescent (GMYC) method, based on the Yule coalescent [25], bPTP, which is a phylogenetic tree method based on the Poisson tree processes [26], BP&P relying on the Markov chain Monte Carlo Bayesian Phylogenetic program [27–29], and the recently proposed ASAP method that employs pairwise genetic distances without phylogenetic inference or prior species information [24]. All these methods have been frequently used as a first step in an integrative taxonomic process, especially to form initial species hypotheses. These methods often yield similar, but distinct results, where some methods (e.g., GMYC and bPTP) have received criticism as tending to overestimate true species richness [21]. Therefore, we focus on the results of somewhat more conservative methods and discuss in the most detail the results from the ASAP analysis. This new method performed well both with simulations [24] and when applied to ´known´ species problems [24], as well as with existing morphological taxonomy in a recent study on old world agama lizards [30] and was among the methods that best aligned with morphology in a recent spider study [23]. We nevertheless emphasize that the results of species delimitation analyses relying only on a single gene cannot provide more than an initial idea of actual species richness, e.g., [23,31].

## 2. Materials and Methods

### 2.1. Study Organisms, Taxon Sampling, and Identification

*Phrynus* are large cryptic arachnids (Figure 1), most active at night and in dark habitats such as under rocks or under/inside/on logs and trees, but typically not in leaf litter or soil. They are abundant in caves and are found actively preying on other arthropods on cave walls at all hours of the day, as well as in total darkness [9,16,17]. *Phrynus* is a New World genus of currently defined 36 extant species restricted to a relatively narrow latitudinal range from southern North America, throughout Central America and the Caribbean, to northern South America, with a single aberrant species from Indonesia [16,17,19,20,32,33]. Approximately 19 described species are restricted to the Central and South American mainland. Cuba has four endemic species, Hispaniola five, Puerto Rico two, Jamaica one, and the Lesser Antilles four. Only *P. marginemaculatus* C.L. Koch, 1841 is thought to be pan-Caribbean ranging from Florida south through multiple islands. We also collected two *Paraphrynus* species, *Pa. cubensis* from Cuba and *Pa. laevifrons,* from South America. The reciprocal monophyly of *Paraphrynus* and *Phrynus* is questionable based on our data, and they may be synonyms. This is unsurprising given the lack of thorough revisionary work and weak diagnosis of the two, relying mostly on the number and length of minor spines on the dorsal margin of the pedipalpal tibia, two versus one in *Paraphrynus* and *Phrynus*, respectively [16,17,19,20,32,33].

This study presents samples from nearly the entire range of the genus, extending across all the Greater Antilles, multiple Lesser Antilles, northern South America, Central America, Mexico, and southern USA (Figure 1, Table 1). The CO1 mitochondrial gene was successfully sequenced from a total of 544 specimens (including [10]), distributed as follows: 23 from Florida; USA (none from caves); 54 (one from one cave) from South America (Colombia); 71 (28 from eight caves) from Mexico and Central America (Guatemala and Costa Rica); 184 (139 from 23 caves) from Puerto Rico; 34 (seven from three caves) from Cuba; 53 (22 from eight caves) from the Dominican Republic; 42 (29 from ten caves) from Jamaica; and 83 (28 from 10 caves) from the Lesser Antilles (see Supplementary Data).

### 2.2. Data

This study employs 544 total specimens including outgroups composed of six specimens from four spiders and two non-*Phrynus* phrynids.

Ingroup specimens are morphologically consistent with the named species *P. alejandroi* (27), *P. barbadensis* (40), *P. damonidaensis* (2 + 4 ´near´ *damonidaensis*), *P. eucharis* (36 + 7 'near' *eucharis*), *P. goesii* (42), *P. longipes* (168), *P. marginemaculatus* (30 + 9 ´near´ *marginemaculatus*), *P. operculatus* (3), *P. pinarensis* (12), *P. pseudoparvulus* (9), *Pa. cubensis* (14), *Pa. laevifrons* (30), and four groups that could not be linked with an existing species: *P.* sp. 1 (24), *P.* sp. 2 (6), *P.* sp. 3 (3), and *P.* sp. 4 (38).

Twelve named species (10 *Phrynus* and two *Paraphrynus*) and 20 'species groups' were initially identified based on morphology using the available taxonomic literature ([19] and literature therein) and blasting sequences against GenBank (https://www.ncbi.nlm.nih.gov/genbank (accessed on 15 February 2023)). Even experts find identification extremely challenging in Phrynidae, and species diagnoses are often weak [14,16–19,32,33]. Our layered approach relied on the literature, aided by the leading taxonomic experts that worked with CarBio (islandbiogeography.org) in the field (A. Pérez González, R. Teruel, C. Viquez) and who assisted us with identifications in the lab (L. Armas pers. comm.).

Specimens were fixed and preserved in 95% ethanol. Genomic DNA was extracted from a single leg using a Quiagen DNeasy Blood and Tissue Kit, eluted in 200 µL of de-ionized $H_2O$, and stored at $-20\,°C$. The CO1 gene was amplified using universal primers following prior protocols (e.g., [34,35]). Amplified fragments were sequenced in both directions by the University of Vermont Cancer Center DNA Analysis Facility within the Vermont Integrative Genomics Resource (VIGR) facility and the University of Arizona Genetic Core, then assembled using the Chromaseq module [36] in Mesquite [37] through Phred and Phrap [38,39], and then proofread. The matrix contained 1192 aligned bases.

Alignment was performed in MAFFT v7.036 [40] using the FFT-NS-i strategy with a 1PAM/k=2 parameter, and a gap opening penalty of 1.53. The resulting alignments were tested for models of substitution in jModelTest2 [41]. Phylogenetic analyses were performed on the unpartitioned matrix. The dataset was analyzed using Bayesian (MrBayes 3.2.1) [42,43] methods using the model suggested by jModelTest 2 (GTR + I+G). Bayesian analyses were run on the CIPRES cluster [44]: 4 runs of 50 million generations. Convergence was assessed with AWTY [45] and the burnin was discarded. A majority-rule consensus tree was constructed from the post-burnin distribution of trees.

### 2.3. Species Delineation and Population Assignment

Four methods were used to delimit the putative species based on CO1 divergences. First, the species limits were searched using the *gmyc* function in the package SPLITS [46] in R [47], based on an ultrametric tree, generated by converting a consensus tree into an ultrametric, dichotomous tree in the R-package ape [48]. The function was used to optimize both the single threshold [49,50] and the multiple-threshold [51] versions of the generalized mixed Yule coalescent which allow for a variable transition from coalescent to speciation among lineages. The latter model was shown in [51] to give a better congruence with classification based on morphology. The analysis was also run by collapsing sequences differing by 3% or less as a single haplotype. Second, a species delimitation analysis used a Bayesian implementation of the Poisson Tree Processes model (bPTP) [26] on the bPTP web server (https://species.h-its.org/ (accessed on 23 September 2021)), which provides tree-based Bayesian support for each of the putative species boundaries. The method is similar to the GMYC method but more flexible as it does not require an ultrametric tree with a known mutation rate [26]. The analysis was run as a rooted tree with outgroups removed for 100,000 generations with 10% burnin removed. Third, the Markov chain Monte Carlo (MCMC) Bayesian Phylogenetics and Phylogeography program (BP&P) was used under the multispecies coalescent (MSC) model [27–29] with joint species delimitation and species tree inference of unguided species delimitation (A11 setting) [29]. Last, species were delimited using the Assemble Species by Automatic Partitioning (ASAP method) run on the ASAP website (https://bioinfo.mnhn.fr/abi/public/asap/ (accessed on 30 January 2023)), based on simple p-distances.

### 2.4. Genetic Distances and Isolation

Estimates of the number of base substitutions per site were calculated using the proportion of different sites between all sequences (raw distance using the R-package ape [48]). The association of the cophenetic distances from the Bayesian tree, within and among the *Phrynus* species, with geographic distances was evaluated with a Mantel test using the R-package vegan [52], permuted 1000 times. Geographic distances were calculated from the geographic coordinates using the command distHaversine in the R-package geosphere [53]. The Mantel test was conducted for all samples and separately for island and mainland samples, and for cave and surface samples.

### 3. Results

Bayesian phylogenetic analyses of our CO1 dataset yielded a tree strongly corresponding to geographical locality. As expected, specimens within localities were often identical, and when using a single gene fragment to group over 500 taxa, some deeper branches of the tree had weak support (Figures 2–5, Supplementary Figures S1–S5, see Section 4). *Paraphrynus* is polyphyletic, with *Pa. laevifrons* (=*Tarantula laevifrons* Pocock, 1894, by original designation), the name-bearing type of the genus [19] nested deep within *Phrynus* (Figure 4), while *Pa. cubensis* is sister to the remaining taxa in this study (Figure 2, clade H, Supplementary Figure S5). The validity of *Paraphrynus* is thus doubtful.

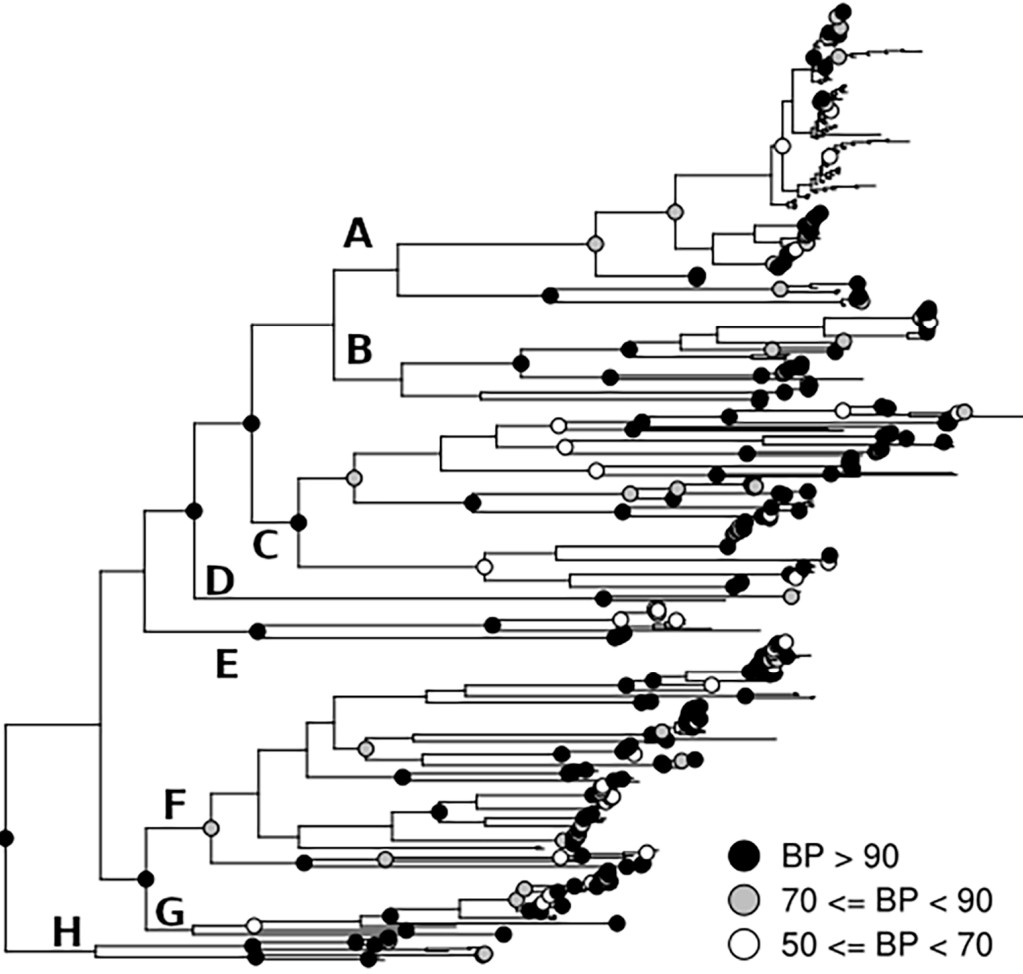

**Figure 2.** Results of phylogenetic analysis of all individuals. Letters on nodes indicate eight subclades; for details, see Figures 3–5 and supplementary materials (Figures S2–S4, Table S1). Circles represent branch support based on Bayesian posterior probability (PB).

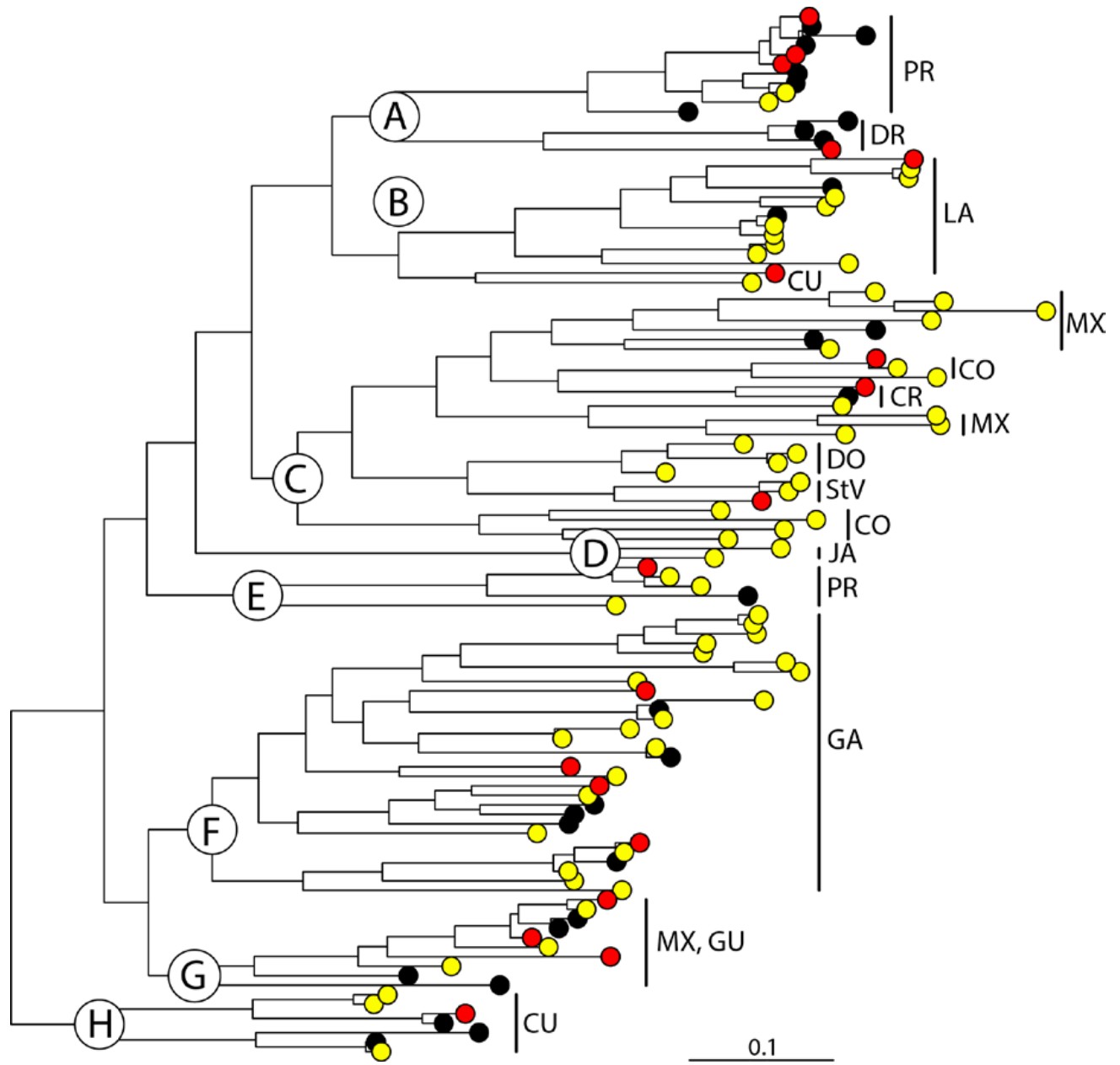

**Figure 3.** Results of phylogenetic analyses showing a single individual from each of the 110 putative species for the PTP analyses. Letters on nodes indicate eight subclades that are detailed in Figures 3–5, Figures S2–S5, and Table S1. Colored circles indicate the habitat of each sampled species; yellow = epigean, black = cave, and red = both. Lines and abbreviations refer to area of origin: PR—Puerto Rico, DR—Dominican Republic (Hispaniola), LA—Lesser Antilles, CU—Cuba, MX—Mexico, CO—Colombia, CR—Costa Rica, DO—Dominica, StV—St. Vincent, JA—Jamaica, GA—Greater Antilles (for detail see Figure 5), GU—Guatemala. Scale bar indicates the scale of branch lengths. Bayesian posterior probability is provided in Figure 1.

Our identification of the species, refined with the aid of the molecular data (BLAST) and phylogenetic structure, constitutes the minimal species diversity hypothesis (Figures 2–5, Table 1). The remaining groups were morphologically found to be more similar to one of these eleven than to other described phrynid species.

Clear divergences were found between populations of mainland and islands, among practically every island, and nearly every isolated cave within each island and mainland.

Species delimitation analyses suggested 66–114 species among the sampled localities. The multiple threshold methods resulted in 92 species (27 singletons). The likelihood of the single method did not deviate from the null model whereas the multiple threshold method was significantly better, with a likelihood ratio of 14.28, *p* < 0.001. When applying the model to the patristic distances obtained from the Bayesian tree, the multiple threshold model resulted in 114 species (35 singletons), *p* =0.001. The single locus bPTP approach [26], criticized by Hedin [54], suggests 110 species (38 singletons), with an average Bayesian support of 0.93 (Supplementary Figure S1). The BP&P analysis agrees with the PTP results and identified the same 110 putative species, and we present trees and further analysis based on these 110 species below. The average divergences within (1.2%) and among (9.4%) these 110 species are generally near, at, or over the so-called barcoding gap threshold ([20], Table 1). All employed delimitation methods agreed on at least 66 putative species, in addition to 27 (up to 37 in PTP analyses) highly divergent singletons for a minimum of 20 species (=species groups) and a relatively conservative estimate of 66+ and up to 114 putative species (Figures 2–5). These findings are consistent with the high species turnover (beta diversity) that characterizes many cave systems [5,7].

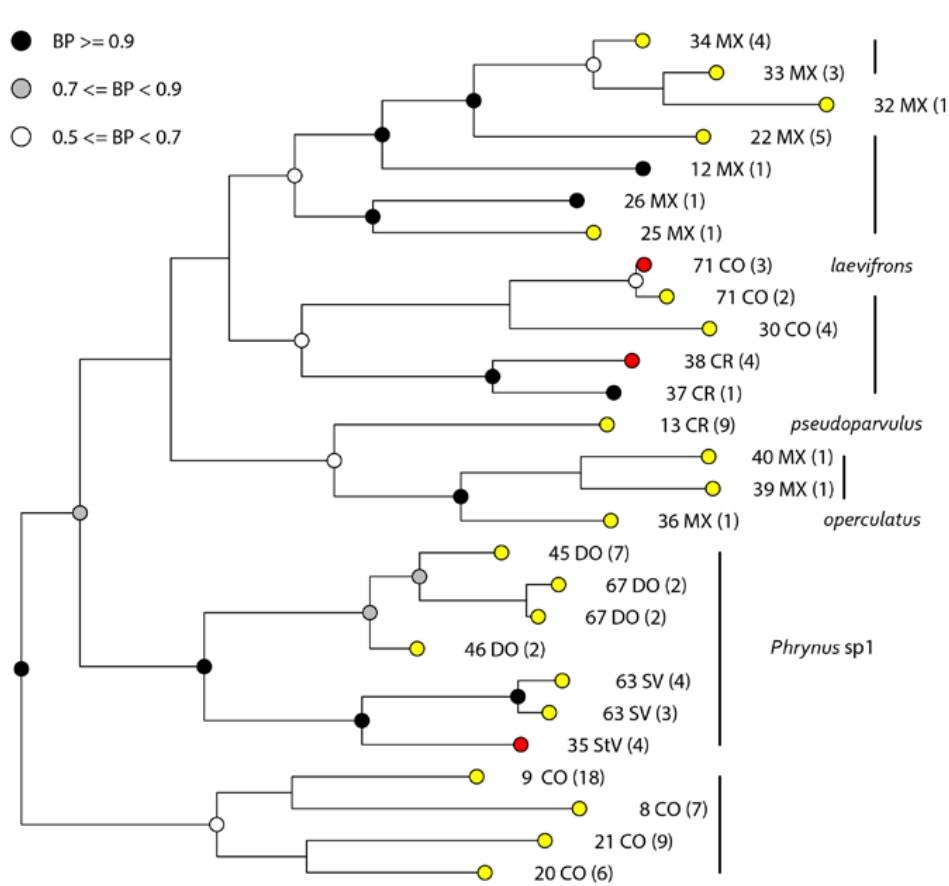

**Figure 4.** Details of clade C from Figure 3. Colored circles indicate the habitat of each sampled species; yellow = epigean, black = cave, and red = both. Abbreviations refer to area of origin: MX—Mexico, CO—Colombia, CR—Costa Rica, DO—Dominica, StV—St. Vincent. Numbers before locality abbreviation refer to ASAP species number (see Table S1), number in parentheses after locality information refer to the number of individuals per putative species. Branches sharing ASAP species numbers indicate additional species implied by the PTP analysis. Lines and names refer to the 20 morphospecies based on existing taxonomy and monophyly in the tree (Figures 1 and 2).

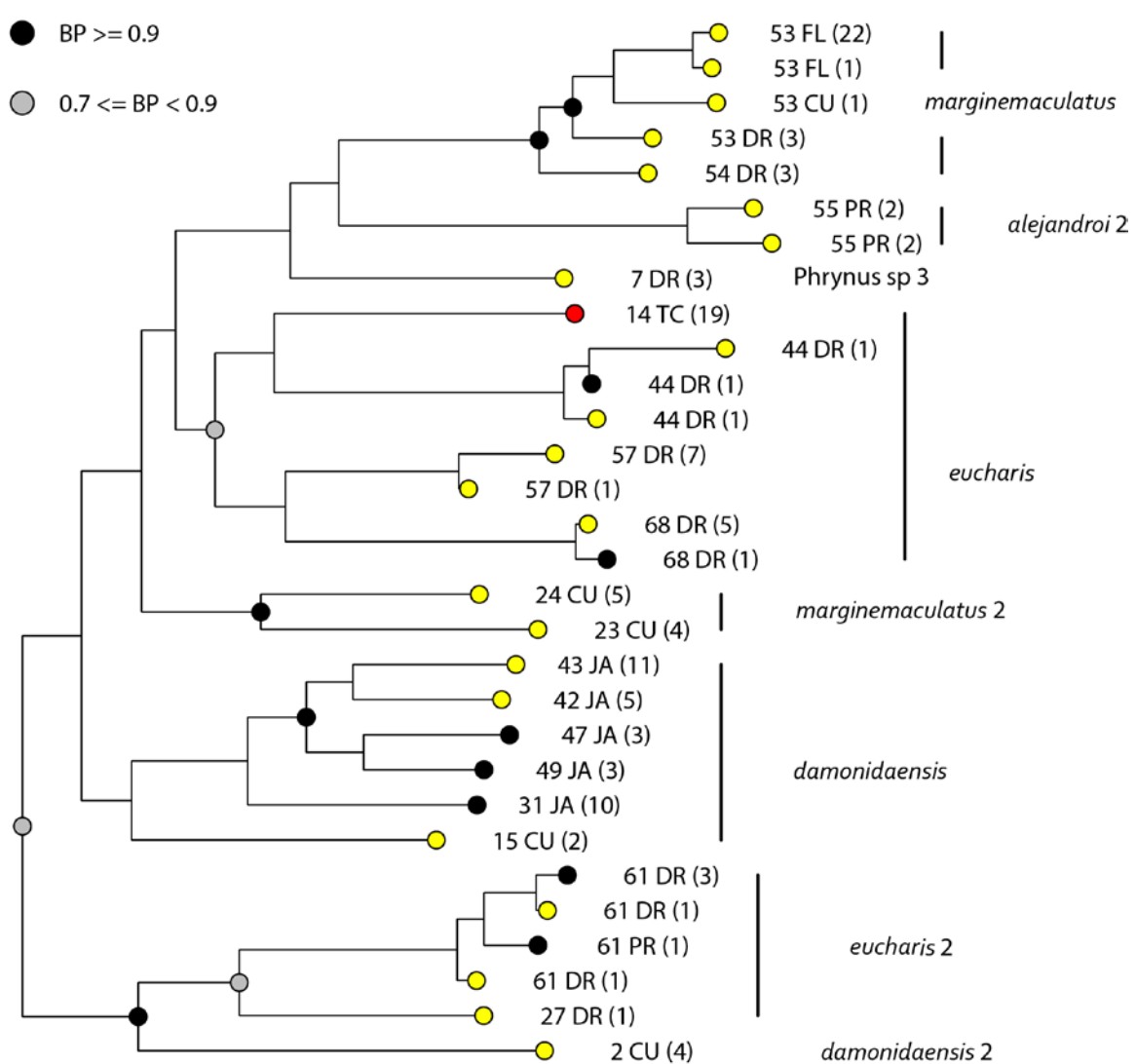

**Figure 5.** Details of clade F from Figure 3. Colored circles indicate habitat of each sampled species; yellow = epigean, black = cave, and red = both. Abbreviations refer to area of origin: FL—Florida, DR—Dominican Republic (Hispaniola), PR—Puerto Rico, TC—Turks and Caicos, CU—Cuba, JA—Jamaica. Numbers before locality abbreviation refer to ASAP species number (see Table S1), number in parentheses after locality information refer to the number of individuals per putative species. Branches sharing ASAP species numbers indicate additional species implied by the PTP analysis. Lines and names refer to the 20 morphospecies based on existing taxonomy and monophyly in the tree (Figures 1 and 2).

The recently developed ASAP method resulted in 73 species, ($p = 9.00 \times 10^{-5}$) and a threshold distance of 0.74. Of the 110 PTP species, 37 clustered together with their closest relatives in clades of two to ten species each with the ASAP (Figures 4 and 5 and supplementary figures). Sixty-five of the ASAP species were restricted to a single country. Five species were found on two islands. One species. *P. marginemaculatus* #55, was found in three countries (the USA, Cuba, and the Dominican Republic).

While a universal cutoff for mtDNA divergences among species is challenging to establish, it often ranges between 3 and 6% in various organisms including

arachnids [49–54]. The 1.2% average within-group distance is much less, although some groups have intragenetic distances above 6%, e.g., *Paraphrynus laevifrons* and *Paraphrynus cubensis*. The most conservative species delimitation analysis, resulting in 66 species, was obtained with GMYC when all sequences differing by 3% or less were treated as a single haplotype. Uncorrected genetic distances in named species (Supplementary Material) ranged up to 7% and among species up to 19%, with the average distance between sequences of 9 % (sd 2.9 %). Geographic distances significantly explained a portion of genetic distances on islands (19%, both among caves and epigean specimens) and mainland epigean specimens (27%). Genetic distances between the cave samples increased overall faster with geographic distance (b = $6.4 \times 10^{-3}$, $p \leq 2 \times 10^{-16}$) than among the epigean samples (b = $4.4 \times 10^{-3}$, $p \leq 2 \times 10^{-16}$). A total of 287 specimens came from epigean habitats, of which 253 were from caves (Figures 2–5, Supplementary materials). Of the maximum 110 putative species, 77 came from islands (specimens) and 33 from the mainland (148 specimens) (Figures 2–5, Table S1, Supplementary material). A total of 14 of the putative 73 ASAP species (19%) were singletons, but approximately 34% were based on the PTP method-.

No species were found on the mainland and on an island, and no species were found in more than one country on the mainland. Only 2 of the 110 putative bPTP species were found on two islands.

In sum, the 12 nominal species may harbor 66 to well over 100 putative species. No putative species-level clades are shared among continents and islands, among islands, or among distant areas within islands, neither epigean nor within caves. Each genetic lineage has a highly restricted range, frequently from a single cave.

## 4. Discussion

Caribbean biodiversity is renowned, yet its greatest dimensions of diversity, both in terms of organisms (e.g., arthropods other than a few select groups) and habitats (especially caves), remain poorly studied. While we celebrate the magnificent biodiversity of the region, we are far from gaining a solid understanding of the Caribbean biota.

We explored the CO1 structure in phrynid species across the Caribbean archipelago and neighboring landmasses through extensive taxon sampling, with emphasis on caves (Figures 1 and 3). A single mtDNA marker is not sufficient to reconstruct a robust phylogeny but does illuminate the mtDNA diversity within and among closely related species such as the Caribbean phrynids in this study. In turn, mtDNA diversity often, though not always, predicts approximate species richness [9,20–23]. Indeed, preliminary evidence from a RAD seq dataset being developed (Jason Bond, James Starrett, I. Agnarsson et al., unpublished) indicates that our fundamental findings are generally supported in analyses of multilocus next-generation sequence data. Therefore, we hypothesize that the observed genetic divergences are indicative of species radiation rather than just mtDNA divergences as a consequence of, for example, stationary females and actively dispersing males. Nevertheless, we are cautious in interpreting the present results.

The placement of *Paraphrynus laevifrons* within *Phrynus* implies that the former is in need of revision, or is possibly a synonymy. This finding is corroborated in forthcoming study using multiple genes [55]. Here, *Paraphrynus* clustered in two clades, where clade H is the sister to the remaining ingroup taxa, while the name-bearing type cluster was in clade C within *Phrynus* (Figures 2–5 and supplementary material: Clades C and H).

The sequences from the mtDNA barcoding gene CO1 imply the split of the 12 named species into many to achieve mutual monophyly. This is not surprising given the likely side fidelity of phrynids, the isolating effects of cave-living, the high number of cave endemics in the region (e.g., [56], Binford et al. unpublished data on *Loxosceles*), the lack of molecular data on phrynids in prior studies, and the challenging taxonomy of phrynids based purely on simple morphological traits, mostly on the apparently unreliable spine structure [14,16–19,31–33,57–62]. For example, individuals morphologically consistent with the ´widespread´ *P. marginemaculatus* are polyphyletic and may represent 3–4 separate

clades. We split the tree, for display purposes, into eight major clades, A-H (Figures 2–5,S1 and S2). A conservative conclusion might recognize 20 phrynid species based on phylogenetic and morphological species concepts. Further scrutiny of the genetic evidence, however, suggests that deep mtDNA divergences have formed among populations within these 20 clades (Figures 1 and 2, Supplementary Materials). These are unique genetic lineages present in every isolated habitat at strongly structured geographical scales (mainland continents vs. islands, among islands, among caves, and some other habitats within islands) (Figures 2–5, Supplementary Materials). The morphological (11) and minimum morphological plus phylogenetic number of species (twenty) likely vastly underestimate the true species richness of the Phrynidae we sampled. Species delimitation analyses indicated 66–114 putative species. We focus our discussion on the ASAP analysis being relatively conservative, and on the likely upper limit of diversity implied by the PTP and the bPTP analyses resulting in 110 putative species. In part, this is arbitrary because our focal arguments would be the same for the range of estimates provided by the species delimitation methods. We feel this approach is practical as it highlights a likely range of actual species richness as even the highest number of estimated species results in a scheme that includes taxa that show high mtDNA average divergences of 9.4% between putative species. This is on par with or greater than typical mtDNA distances among related arachnid species (e.g., [21,63]). Comparable divergences are typically found between, not within species of arachnids, especially in lineages that have been carefully revised using morphology and DNA analyses. Even our most liberal estimate of species seems plausible. However, we emphasize that this DNA barcode analysis is an initial assessment of mtDNA and Caribbean *Phrynus* species diversity, to be further examined by genome-wide sequence data combined with detailed morphology, e.g., [64]. Because the scope, density, and narrow endemicity of Caribbean Phrynidae is, frankly, incredible, mating trials may be required to prove the species' status.

Despite the potentially extensive diversity discovered, our sampling was modest. We have only explored approximately 63 caves out of the more than 7000 in the Caribbean. We only sampled 17 out of over 100 islands and only a few regions and habitats within the islands. Thousands of genetically unique lineages likely exist in unexplored cave systems, habitats, and islands. The total Caribbean phrynid species richness may be two orders of magnitude greater than currently known. If other taxa show similar short-range endemism and morphologically cryptic species ([65], Binford et al., unpublished data), the current estimates of Caribbean biodiversity are far too low, especially for cave-liking taxa. Although we know only a fraction of the diversity in world hot spots, this study documents a multiplication factor of impressive power and importance. In this context, it is worth noting that a total of 15 described species have been recorded from the Caribbean, and many of those were not sampled in our study. We cannot rule out misidentification given the weak taxonomy; however, many *Phrynus* species have very narrow ranges, some only found at their type localities [19]. Hence, the more likely explanation of why we failed to sample some known Caribbean species is that we did not specifically visit type localities.

Landmass (continents and islands) lineages were found to be deeply genetically divergent, and cave system and some epigean region lineages were slightly less so. Sampling from epigean and cave habitats was fairly even in our study. Of the maximum 110 putative species, 46 were from caves, 29 exclusively. This supports our hypothesis that caves may promote speciation in lineages that live both in cave and epigean habitats. A total of 37 of the 110 putative species (34%) were singletons compared to approximately 30–35% of singletons found characteristic for biodiversity surveys of tropical arthropods [66]. A high number of singletons generally implies an undersampling bias. There is no doubt that hundreds or thousands of putative mtDNA species remain to be discovered in the Caribbean [9].

Geographical distance is the obvious explanation for genetic divergences (see also [67]). For example, only two species were both found on the mainland (Florida) and on an island. Further, no species were found in more than one country on the mainland, and only 5 of

the 73 putative ASAP species were found on two islands. Geographic distance significantly explains variation in the genetic distance in Caribbean phrynids both in epigean and cave habitats on islands, and among epigean mainland habitats. Nevertheless, geographic distance does not explain genetic distances among mainland caves, which on average accounts for less than 20% of genetic distances among intra-island habitats and 27% among mainland epigean habitats. Phrynids in the Caribbean seem to be diverging and speciating rapidly due to geographic distances and other important factors. High site fidelity and distance-independent environmental barriers may help explain this diversification. That nearly all putative species discussed here are of very short-range cave—or single epigean locality—endemics suggests that the cave lifestyle plays an important role.

Can the relative importance of different geographic scales of diversification be estimated? Did most speciation occur among different landmasses (across oceanic barriers) or within landmasses among caves? The precision of such estimates is limited by the available data; we sampled islands more intensively than caves. The large proportion of single-cave endemics indicates that small-scale diversification is a major driver of phrynid speciation. Caves may be speciation ´turbo engines´ and cave lineages may be more diverse than comparable lineages only found in epigean habitats [9].

## 5. Conclusions

For the first time, we assessed the diversity of *Phrynus* and *Paraphrynus* across much of their natural range. Sampling within the Caribbean archipelago was especially strong. mtDNA divergences suggest short-range endemism and high beta diversity. The data suggest that the eleven currently named species could balloon to 66–114 ´barcoding species. While further data are necessary to demonstrate that the distinct genetic clusters represent phylogenetic and/or biological species, we have shown that *Phrynus* mtDNA lineages have massively diversified in the Caribbean. A prior study [9] identified this trend but explored only three islands. This larger and denser study shows that Caribbean Phrynidae are much more diverse than hitherto appreciated, which is likely explained by their isolated cave habitats. We predict similar results for other cave-liking lineages. Our findings significantly support the importance of the Caribbean as a world hotspot of biodiversity.

**Supplementary Materials:** The following supporting information can be downloaded at: https://www.mdpi.com/article/10.3390/taxonomy3010011/s1. Table S1: Specimens used in analysis, specimen codes, collecting localities, population/species group assignments from analysis]. Stars indicate specimens from epigean habitats; Figure S1: Results of the Bayesian analysis of all data including specimen detail omitted from Figures 1 and 2; Figure S2. Details of clade A from Figure 3. Colored circles indicate habitat of each sampled species, yellow = epigean, black = cave, and red = both. Abbreviations refer to area of origin: PR—Puerto Rico, DR—Dominican Republic (Hispaniola). Numbers before locality abbreviation refer to PTP species number (see Table S1), number after locality information refer to number of individuals per species; Figure S3: Details of clade B from Figure 3. Colored circles indicate habitat of each sampled species, yellow = epigean, black = cave, and red = both. Abbreviations refer to area of origin: StK—St. Kitts, JA—Jamaica, StE—ST. Eustatius, BA, Barbuda, StB—St Barts, StM—St Martins, AN—Antigua, MO—Mona, PR—Puerto Rico, GU—Guadalupe, BS–Barbados, CU—Cuba. Numbers before locality abbreviation refer to PTP species number (see Table S1), number after locality information refer to number of individuals per species; Figure S4: Details of clades D and E from Figure 3. Colored circles indicate habitat of each sampled species, yellow = epigean, black = cave, and red = both. Abbreviations refer to area of origin: JA—Jamaica, PR—Puerto Rico AN—Antigua, CO—Colombia. Numbers before locality abbreviation refer to PTP species number (see Table S1), number after locality information refer to number of individuals per species; Figure S5: Details of clades G and H from Figure 3. Colored circles indicate habitat of each sampled species, yellow = epigean, black = cave, and red = both. Abbreviations refer to area of origin: MX—Mexico, GT—Guatemala, CU—Cuba. Numbers before locality abbreviation refer to PTP species number (see Table S1), number after locality information refer to number of individuals per species.

**Author Contributions:** I.A. and L.C.-Q. designed the study, I.A. and L.C.-Q. conducted fieldwork, L.C.-Q. and L.J.M.-C. performed the molecular work, I.A. and S.P. analyzed the data and produced Figures, I.A. and J.A.C. acquired the funding, I.A. wrote a draft of the manuscript, and all authors contributed to rewriting and editing. All authors have read and agreed to the published version of the manuscript.

**Funding:** This research is supported by NSF DEB-1050187, 1050253, and 1314749 to IA and GB, NSF DBI-1349205 to D. Barrington, I. Agnarsson, and CW Kilpatrick, NSF DBI-1003087 to LE, a National Geographic WW-203R-17 grant to I. Agnarsson, and the Global Genome Initiative, Smithsonian institution, to JAC. This work represents the collective effort of team CarBio and we are thankful to all the participants in this project. We especially thank Lauren Esposito, Rolando Teruel, Abel Pérez Gonzáles, and other members of the team that have passion for Amblypygi. Some of this work was performed while L. Caicedo-Quiroga worked for the Walter Reed Biosystematics Unit and was performed under a Memorandum of Understanding between the Walter Reed Army Institute of Research and the Smithsonian Institution. The published material has been reviewed by the Walter Reed Army Institute of Research. There is no objection to its presentation and/or publication. The opinions or assertions contained in this study are the private views of the authors and are not to be construed as official or as reflecting true views of the Department of the Army or the Department of Defense.

**Data Availability Statement:** All molecular data were submitted to GenBank: accession codes OQ450552 - OQ451095, SUB12837539. Data matrices and trees will be submitted to Dryad and are available from the corresponding author.

**Acknowledgments:** Special thanks to the Sociedad Espeleológica de Puerto Rico and Sociedad Espeleológica Unida del Sur for assistance with fieldwork and various logistics. Furthermore, thanks to Departamento de Recursos Naturales y Ambientales de Puerto Rico for arranging permits and to Isla de Mona park rangers, and especially Tony Nieves for valuable help in the field. Additional logistic support was provided by Fideicomiso de Conservación de Puerto Rico, Universidad Interamericana de Puerto Rico, and Casa Verde, Maunabo. Greta Binford was instrumental in securing funding for this study and participated in fieldwork and conceptual framework of this and other papers from CarBio. Kenneth J. Chapin kindly provided CO1 sequences of several specimens from Puerto Rico, as well as specimens from the United States. Yadira Ortiz-Ruiz, Emma Chereskin, Margaretta Kuhn, and Alexis Avonda helped with DNA extraction and PCRs of a portion of the specimens here included.

**Conflicts of Interest:** The authors report no conflict of interest.

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
