# Peer review of "Deep mtDNA Sequence Divergences and Possible Species Radiation of Whip Spiders (Arachnida, Amblypygi, Phrynidae, Phrynus/Paraphrynus) among Caribbean Oceanic and Cave Islands"

_2673-6500, doi:10.3390/taxonomy3010011_

Round 1
Reviewer 1 Report
The manuscript is a very important contribution about the the diversification of arachnids from a very biodiverse region of the Americas. However, there are some part of the manuscript thagt must to be improved previously to be accepted (see comments on the notes into the PDF version attached to this email):
1. In the Introduction, I strongly recommend to add something about the species delimitation methods in arachnids, there are many papers published last years with spiders about this topic, the introductions is quite short for this important topic.
2. There are some inconsistences in information into the Introduction section, tsome phrases/paraghraphs must to be in the Material and Methods section, not in the introduction.
3. The authors must to be careful with some aseverations, such as the monophyly of Phrynys and Paraprhynus is not suported with their data, however they used only one molecular marker, besides this molecular marker is CO1 and more genes including nuclear markers are necessary to strongly support this idea.
4. Figure 1 has poor quality and there is too much space in the right side of the figure.
5. Line 107: Mexico belongs to North America, not Central America. Very common mistake.
6. The use of the outgroups is not well explained , were they used to test monophylo of something?, the topologies show the the outgroups (labeled as DR-Domicican Rep.) even are grouped ionto the internal groups, why?, please explain with detail.
7. line 30, about the phylogenetic analyses, due to CO1 is a codificant gene, why did not use the third position of the codon? this explanations is missing.
8. Line 169, this is Discussion not Results. see the notes into the PDF file.
9. In the supplementary material, the authors mentioned that teh Gen Bank accessions numbers are in progrees,?, I strlongy recommend that tha authors included a Table or appendage showings these numbers. It is mandatory that the authors included all the information regarding the manuscript.
10. Line 179: this is not Results, this is M&M.
11. In the topologies the branch support values are missing... I strongly recommend that at least the general topology includes the support values on the branch, this is very important in any phylogenetic reconstruction analysis.
12. The quality of the Figures must to be improved.
13. Line 226: to speak of monophyly, it is important to add the branch support values.
14. Because you are using CO1, I strongly recommend to include an haplotype network to calculate the number of mutations per haplogroup or haplotype.
15. Finally, a discussion about the different species delimitation methdos is neccesary, there are some papers working with species delimitation methods with tarantulas and brown recluse spiders from Mexico and North America where the advantages and disadvantages are discussed on such as methods, based on the molecular markers used commonly. An additional of literature research is nedeed.

Reviewer 2 Report
The manuscript entitled “Deep mtDNA sequence divergences and possible species radiation of whip spiders (Arachnida, Amblypygi, Phrynidae, Phrynus/Paraphrynus) among Caribbean oceanic and cave islands” presents an interesting contribution on the Caribbean diversity of whip spiders. The manuscript is well written and I suggest you to have it published after some minor corrections that I report below:
Line 138: GMYC requires an ultrametric tree, when and how do you generate it?
Line 188: all the species delimitetion models that you have used are based on coalescent theory, so maybe a so hight number of species rappresented only by one speciments could affect the analysis. Have you tried to do the analysis also after removed the singletons?
Round 2
Reviewer 1 Report
I dont have further revisions, the paper can be published.